# Investigating the resistome, taxonomic composition, and mobilome of bacterial communities in hospital wastewaters of Metro Manila using a shotgun metagenomics approach

Jiaan Carlo E. Santos,[1,2] Daphne Janelyn L. Go,[2,3,4] Robert D. Unciano,[2,5] Paul K. Yu,[6] Angelyn R. Lao,[2,7] Ma. Luisa D. Enriquez,[1,2,8] Llewelyn M. Espiritu,[1,2] Anish M. S. Shrestha[2,3,4]

**ABSTRACT**   We profiled antibiotic resistance genes, bacterial communities, and mobile genetic elements in untreated hospital wastewater from three tertiary hospitals in Metro Manila using shotgun metagenomic sequencing. The resistome analysis revealed high abundances of genes known to confer resistance against sulfonamides (*sul1*, *sul2*), aminoglycosides (*aadS*), and macrolides/streptogramins (*msrE*, *mphE*). High-risk resistance genes were also detected, including those known to confer resistance to β-lactams ($bla_{OXA}$, $bla_{TEM}$, $bla_{GES}$, $bla_{NDM}$, $bla_{KPC}$), colistins (*mcr-5*), and tetracyclines [*tet(C)*, *tet(A)*, *tet(L)*, *tet(M)*]. Comparisons with hospital wastewater resistome profiles from regional neighbors and other lower-and-middle-income countries indicated broadly similar relative abundances of dominant resistance genes, with differences largely driven by low-abundance resistance genes. The bacterial community was dominated by the phylum Pseudomonadota, with high relative abundances of the genera *Stenotrophomonas*, *Rhodococcus*, and *Pseudomonas*, while ESKAPEE pathogens were detected at lower levels. A diverse array of mobile genetic elements—many known to be associated with resistance to multiple drug classes and disinfectants—was also observed. Overall, this study provides a valuable preliminary evidence base for future antimicrobial resistance and epidemiological surveillance efforts in the Philippines, particularly those employing wastewater-based approaches.

**IMPORTANCE**   Antimicrobial resistance (AMR) is a growing public health threat caused by pathogenic bacteria that are no longer controlled by commonly used treatments. Infections caused by these resistant bacteria may lead to prolonged illness, more severe symptoms, or even death. Hospitals are critical hotspots for the emergence and spread of AMR. Their wastewater, which contains antibiotics, medical and human waste, and diverse microbial communities, can support the persistence and dissemination of resistant bacteria. The significance of this research lies in identifying and characterizing these bacterial communities and the resistance genes they carry. Such information can provide an indication of the resistance burden faced by patients and serve as an early warning system to strengthen infection prevention and control measures, support national surveillance efforts, and inform the development of more effective treatment and management strategies in healthcare settings.

**KEYWORDS**   metagenomics, antibiotic resistance, bioinformatics, hospital wastewater, resistome

A ntimicrobial resistance (AMR) is one of the most pressing global health threats. In 2019, an estimated 1.27 million deaths were directly attributed to bacterial AMR and 4.9 million deaths associated with it (1). It is predicted that the economic burden

Address correspondence to Anish M. S. Shrestha, anish.shrestha@dlsu.edu.ph.

The authors declare no conflict of interest.

See the funding table on p. 14.

of AMR-related healthcare costs will reach USD 1 trillion annually by 2050 (2). The Philippines, a lower-middle-income country, is among those disproportionately affected, with an estimated 15,700 deaths attributed directly to resistant infections and 56,700 deaths associated with AMR in 2019 (1). Effective surveillance methods are critical for tracing and understanding AMR risks, including the spread of antibiotic resistance genes (ARGs), discovery of novel resistance mechanisms, and identification of bacterial pathogens carrying these genes (1, 3). The Philippine Department of Health established the Antimicrobial Resistance Surveillance Program (ARSP) in 1988 to monitor clinically significant antibiotic-resistant bacteria (ARB) from clinical samples. This program has reported an alarming increase in AMR in recent years. In 2023, the program documented rising resistance trends, for example, to carbapenems in *Escherichia coli* and *Klebsiella pneumoniae*; to ceftazidime, piperacillin, carbapenems, and possible multidrug resistance in *Pseudomonas aeruginosa*; and to ceftazidime, carbapenems, piperacillin, and cotrimoxazole in *Acinetobacter baumannii* (4).

Laboratories participating in the ARSP utilize culture-dependent methods for antibiotic susceptibility testing on clinically significant species. However, there remains a need for more comprehensive approaches in addressing the full scope of AMR. Shotgun metagenomics provides a powerful approach to analyze the genetic material of microorganisms, including unculturable bacteria, eliminating the biases inherent in culture-dependent methods (5–7). This approach enables comprehensive identification and profiling of microorganisms, ARGs, and mobile genetic elements (MGEs) that mediate transfer and spread of resistance genes.

Hospitals are recognized hotspots for AMR due to the frequent use of antibiotics in treating several pathogens. Hospital wastewater (HWW), in particular, contains a variety of waste materials, including human waste, medical waste, hazardous chemicals, and antibiotics originating from diverse hospital-related activities (5, 8, 9). This environment fosters a high concentration and diversity of ARGs and bacterial pathogens, promoting mutations and horizontal gene transfer (HGT) between ARBs, making HWW an important medium for capturing a snapshot of AMR activity (10). The presence of ARGs in HWW can reflect hospital-associated resistance pressures arising from patients, healthcare workers, hospital infrastructure, and routine clinical and non-clinical activities, providing insight into the AMR burden within the hospital environment. Indeed, several studies worldwide have applied shotgun metagenomics for AMR profiling in HWW (see, e.g., references 7, 11–14).

The objective of this exploratory study is to characterize the ARG abundance profile, bacterial community composition, and the occurrence of MGEs in untreated hospital wastewater (HWW) from tertiary hospitals in Metro Manila, Philippines, using whole-genome shotgun metagenomics. To the best of our knowledge, this work represents the first metagenomics-based profiling of HWW in the Philippines and establishes an initial evidence based on the diversity and prevalence of ARGs and clinically relevant pathogens in healthcare-associated wastewater, thereby informing future strategies for antimicrobial resistance surveillance and mitigation in the country.

## MATERIALS AND METHODS

### HWW sampling

HWW samples were collected biweekly over a 3-month period (May to July 2023) between 8:00 and 11:00 AM from three hospitals (Hospital A, Hospital B, and Hospital C) located in Metro Manila. Untreated HWW samples were obtained from the collection points immediately prior to wastewater treatment, using sterile collection bottles with a total volume of 1,000 mL. Samples A1–A6 were collected from Hospital A from a collection pipe right before the primary collection tank of the treatment facility. Samples B1–B6 were collected from hospital B from a manhole directly before the primary collection tank of hospital B's treatment facility. Samples C1–C6 were collected from hospital C from a collection chute right before the primary treatment tank of their

treatment facility. Each hospital employs its own treatment process, which includes physicochemical and biological methods, before discharging the wastewater into the city sewage system. Sample information is summarized in Table 1.

The surface of the sample bottles was sterilized with ethyl alcohol, placed in a zip bag, labeled, transferred on ice (2°C–4°C), and processed within 5 h after collection. The collected HWW samples were subjected to vacuum filtration using 0.22 µm polyethersulfone (PES; Merck Millipore) membrane filters connected to a vacuum filtration assembly (AGAP Laboratories) powered by a portable vacuum pump. For each sample, 100–150 mL of HWW was filtered until clogging of the membrane occurred. The used filters were then transferred into 15-mL sterile conical centrifuge tubes (Falcon) and stored at 2°C–4°C until DNA extraction.

## DNA extraction

Bacterial DNA was extracted using the DNeasy PowerWater extraction kit (Qiagen, 2020) per the manufacturer's protocol. The quality and quantity of the extracted DNA were evaluated using a NanoDrop 2000c spectrophotometer, yielding >5 ng/µL concentration, purity ratios (A260/280) >1.8, and volumes >10 µL. DNA integrity was checked via electrophoresis (Vivantis) on a 1.2% (wt/vol) agarose gel run at 100 V for 30 min. All samples met the quality control criteria for high-quality DNA extracts. The DNA samples were resuspended in an elution buffer and stored at −80°C prior to sequencing. For sequencing, 20 µL aliquots of each sample were placed in 2-mL sterile microcentrifuge tubes sealed with parafilm and submitted for sequencing.

## DNA sequencing

A total of 18 DNA extracts were subjected to Nextera XT library preparation and shotgun metagenomic sequencing using the Illumina Novaseq6000 platform at the Macrogen Laboratories, South Korea. Shotgun metagenomic sequencing generated an average of 8.8 Gbp of raw sequence data per sample with a read length of 150-bp paired-end reads.

## Quality control of sequence data

Quality assessment of the raw sequence data was performed using FastQC (v 0.12.1) (15) and MultiQC (v 1.14) (16). Trimming of adapter sequences, primers, and low-quality reads

**TABLE 1** Sample information

| Sample ID | Hospital | Date of collection | Type of hospital | Private or public | Number of beds available |
|---|---|---|---|---|---|
| A1 | A | May 2023 | Tertiary | Private | 650 |
| A2 | A | May 2023 | Tertiary | Private | 650 |
| A3 | A | June 2023 | Tertiary | Private | 650 |
| A4 | A | June 2023 | Tertiary | Private | 650 |
| A5 | A | July 2023 | Tertiary | Private | 650 |
| A6 | A | July 2023 | Tertiary | Private | 650 |
| B1 | B | May 2023 | Tertiary | Private | 800 |
| B2 | B | May 2023 | Tertiary | Private | 800 |
| B3 | B | June 2023 | Tertiary | Private | 800 |
| B4 | B | June 2023 | Tertiary | Private | 800 |
| B5 | B | July 2023 | Tertiary | Private | 800 |
| B6 | B | July 2023 | Tertiary | Private | 800 |
| C1 | C | May 2023 | Tertiary | Private | 812 |
| C2 | C | May 2023 | Tertiary | Private | 812 |
| C3 | C | June 2023 | Tertiary | Private | 812 |
| C4 | C | June 2023 | Tertiary | Private | 812 |
| C5 | C | July 2023 | Tertiary | Private | 812 |
| C6 | C | July 2023 | Tertiary | Private | 812 |

was performed using Cutadapt (v 4.2) (17), with parameters provided by the sequencing facility based on the Nextera XT library preparation protocol. The following settings were used: -a CTGTCTCTTATACACATCT -A CTGTCTCTTATACACATCT for forward and reverse adapter sequences, respectively, -O 10 to remove adapters with a minimum overlap of 10 bases, -m 30 to discard reads shorter than 30 bases, and -q 20 to trim bases with a quality score below 20.

## ARG identification and quantification

Bowtie2 (v 2.5.1) (18) was used to align the sequencing reads against the Comprehensive Antibiotic Resistance Database (CARD) (v 3.2.6, accessed June 2023) (19). We selected the CARD as the reference ARG database due to its extensive coverage of resistance determinants and associated antibiotics, and its structured antibiotic resistance ontology (ARO) for drug-class annotation. Only nucleotide reference sequences from the CARD Protein Homolog database (nucleotide_fasta_protein_homolog_model.fasta) were used for alignment; resistome- and variant-based databases were excluded. The alignment was performed using the command bowtie2 -x CARD-db −1 reads1 −2 reads2 -D 20 -R 3 -N 1 -L 20 -i S,1,0.50, which corresponds to Bowtie2's very sensitive preset.

For an ARG gene $G$, its number of copies per genome in a sample (20) was estimated as follows:

$$n_G = \frac{150 \times r_G / l_G}{B \times \text{average bacterial genome size}},\tag{1}$$

where $r_G$ is the number of reads mapped to $G$, $l_G$ is the nucleotide length of $G$, $B$ is the number of bases in the sample, and 150 refers to the length of the sequence reads. The relative abundance of $G$ in that sample, denoted by $a_G$, was then computed using:

$$a_G = \frac{n_G}{\sum_H n_H},\tag{2}$$

where the summation is across all ARG $H$ in the sample.

All resistance genes in the CARD database were clustered into their corresponding drug classes. These drug classes categorize ARGs based on their mechanisms of action and the bacterial pathways they target. The clustering was done using the ARO numbers associated with each resistance gene. For a sample, the total count of ARGs belonging to a drug class $D$, denoted by $n_D$, was calculated using equation 3.

$$n_D = \sum_{G \in D} n_G\tag{3}$$

The relative abundance of $D$ in that sample, denoted by $a_D$, was then computed using:

$$a_D = \frac{n_D}{\sum_T n_T}\tag{4}$$

where the summation is across all drug classes I in the sample.

## Bacterial taxonomy identification and quantification

Taxonomic identification of bacterial communities was performed with Kraken 2 (21) using Standard Kraken 2 16 GB Database. The relative abundances of various microbial genera were estimated using Bracken (22).

## MGE identification and quantification

MGEs were identified by aligning the reads to three reference databases: ISFinder database for insertion sequences (IS) (23), PlasmidFinder database for plasmid replicons (24), and INTEGRALL database for integrases (25). The alignment was performed using Bowtie2 (v2.5.1) with its very sensitive preset: bowtie2 -x database_index −1 reads1 −2 reads2 -D 20 -R 3 -N 1 -L 20 -i S,1,0.50, where database_index corresponds to the Bowtie2 index generated from the ISFinder, PlasmidFinder, or INTEGRALL databases.

The relative abundance of an IS $I$ in a sample, denoted by $a_I$ was computed using:

$$a_I = \frac{r_I/l_I}{\sum_H r_H/l_H},\tag{5}$$

where $r_I$ is the number of reads mapped to $I$, $l_I$ is the nucleotide length of $I$, and the summation is across all insertion sequences $H$ in the sample.

The relative abundances of plasmid replicons in PlasmidFinder and integrases in INTEGRALL were computed similarly.

## RESULTS AND DISCUSSION

### *sul1*, *sul2*, *aadS*, *msrE*, *mphE*, *qacEΔ1*, and *qacL* were the predominant ARGs across all samples

We observed a wide diversity of ARGs across all our samples. Figure 1 shows the relative abundances of the top ARGs found at each hospital. The hospitals exhibited similar sets of high-abundance ARGs, which include *sul1*, *sul2*, *aadS*, *msrE*, *mphE*, *qacEΔ1*, and *qacL*.

Of these, *sul1* and *sul2* are known to confer sulfonamide resistance by encoding alternative dihydropteroate synthase (DHPS) enzymes that reduce antibiotic efficacy (26). Previous research has frequently identified these plasmid-mediated genes in clinical isolates belonging to the genera *Acinetobacter*, *Escherichia*, *Klebsiella*, *Stenotrophomonas*, and *Staphylococcus* (19). The prevalence of *sul* genes has been documented globally, including in lower and middle-income countries (LMICs), such as South Africa and India, where they are predominant in HWW (6, 14).

The streptomycin resistance gene *aadS* encodes an aminoglycoside acetyltransferase that inactivates aminoglycoside antibiotics (19, 27). This gene is typically phenotypically silent in wild-type *Bacteroides*, with expression induced only by a chromosomal mutation (28). Previous studies have shown that *aadS*, which is associated with the phylum Bacteroidota, is commonly located on insertion sequences and transposons, indicating its potential role in horizontal gene transfer. (29, 30). Its presence has been documented in wastewater systems, including untreated HWWs from South Korea, Mexico, and several member states of the European Union (31–33).

The *msrE* and *mphE* genes encode an ABC-F subfamily protein and a macrolide phosphotransferase, respectively, which protect against and inactivate streptogramin

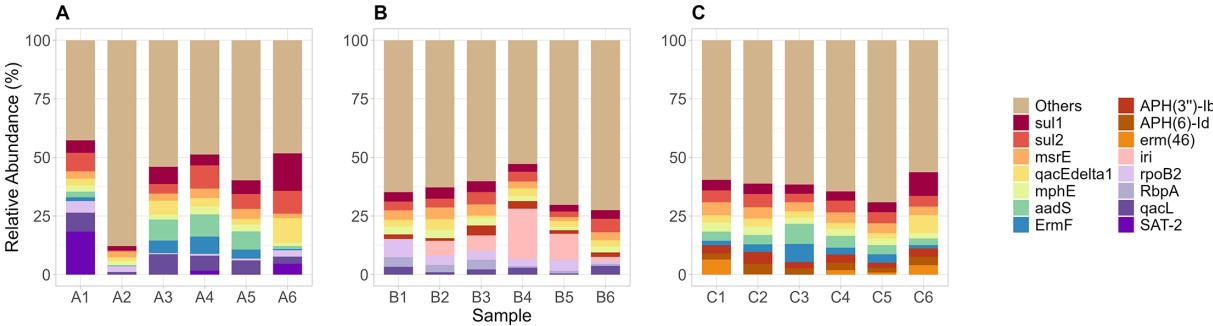

**FIG 1** Relative abundances of the top ARGs from hospital A (A), Hospital B (B), and Hospital C (C). ARGs not part of the top were grouped as "others."

and macrolide antibiotics. Studies have reported that both genes confer higher levels of resistance against second-generation macrolides compared to first-generation macrolides, underscoring the potential influence of increased clinical use of second-generation macrolides in recent years (34, 35). Studies have reported that both ARGs are frequently associated with *Acinetobacter*, *Pseudomonas*, *Escherichia*, and *Klebsiella*, among other genera (19, 36). The prevalence of both genes has been documented in HWWs across various countries, including the Czech Republic and Scotland (UK) (11, 13)

The *qacEΔ1* and *qacL* genes confer resistance to quaternary ammonium compounds (QACs) widely used as surfactants, antimicrobials in disinfectants, and cleaning products (37). Both genes have been associated with *Acinetobacter*, *Klebsiella*, *Pseudomonas*, and *Stenotrophomonas*, among others (19, 38). Prior research has demonstrated that both ARGs are typically plasmid-encoded, thereby increasing their potential for horizontal gene transfer and environmental dissemination (39). Their presence has been documented in HWWs from several countries, including China, Israel, and South Africa (6, 8, 40, 41).

We note an important caveat that the ARGs we identified here are inferred from read-mapping counts. As these genes comprise diverse variants with heterogeneous resistance phenotypes, their detection does not necessarily indicate clinically relevant resistance in all instances. Also, we only detect DNA, which does not necessarily imply gene expression or phenotypic resistance and thus does not directly translate into clinical risk.

## Aminoglycoside, macrolide, β-lactam, tetracycline, and sulfonamide-associated ARGs had the highest combined relative abundances

We consolidated the relative abundances of the ARGs based on the drug classes they are reported to be associated with in the CARD database, as shown in equation 4. Figure 2 shows the top 10 drug classes with the consolidated abundances. The most abundant consolidated drug classes include aminoglycosides, macrolides, β-lactams, tetracyclines, and sulfonamides.

Among these, aminoglycosides are commonly used to treat severe Gram-negative bacterial infections, acting by binding to the 30S ribosomal subunit and inhibiting protein synthesis. Resistance is often mediated by ARGs encoding aminoglycoside-modifying enzymes (AMEs), including *aph(3ʺ)-Ib, aph(6)-Id, aac(6ʹ)-Ib7, aac(6ʹ)-IIa,* and *aac(6ʹ)-II,* which were observed in our samples. These ARGs have been found to be plasmid-mediated, increasing the likelihood of HGT across bacterial species (42–45). The WHO classifies aminoglycosides as critically important antimicrobials, highlighting the urgent need to preserve their clinical efficacy and prevent the potential spread of resistance (3).

Macrolides are commonly used to treat pneumonia and manage cystic fibrosis infections caused by multidrug-resistant, hospital-acquired *Pseudomonas aeruginosa* (46). Multiple ARGs conferring resistance to macrolides were detected in our samples, including efflux pump genes (*smeE, mefC, axyY, mexK, smeD*), ribosomal target modification genes (*erm(46)*), and antibiotic inactivation genes (*mphG, ereD, eraA2,*

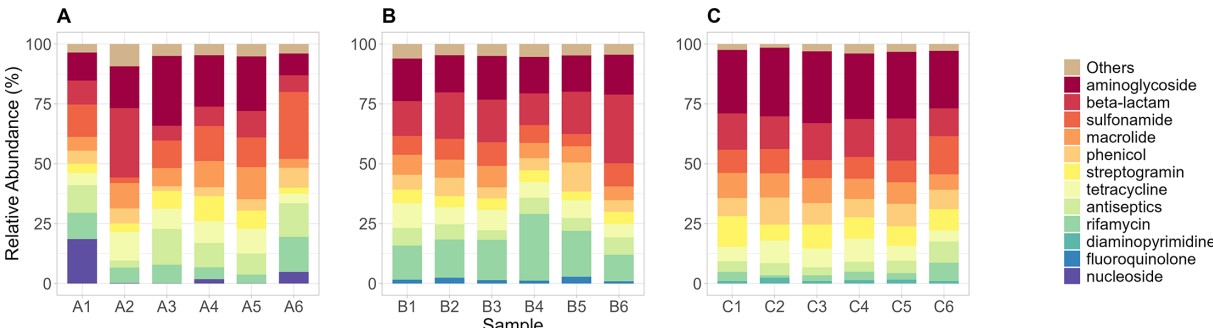

**FIG 2** Top 10 drug classes based on combined relative abundances of associated ARGs identified in Hospital A (A), Hospital B (B), and Hospital C (C). Drug classes not part of the top 10 were grouped as "others."

*mphF*). The widespread use of macrolides in both clinical and agricultural settings has contributed to the increasing prevalence of resistance among pathogenic organisms (47–49). Like aminoglycosides, macrolides are classified by the WHO as critically important antimicrobials (3).

β-Lactam antibiotics are among the most widely used agents to combat a broad spectrum of Gram-positive and Gram-negative bacterial infections. They act by inhibiting cell wall synthesis through binding to essential penicillin-binding proteins (50). β-Lactams, particularly 3rd- and 4th-generation cephalosporins, are classified as highest priority critically important antimicrobials, as their proper use is essential to preserve effectiveness against the increasing threat of multidrug-resistant *Enterobacterales* worldwide (3). We detected $bla_{OXA}$, $bla_{GES}$, $bla_{TEM}$, $bla_{KPC}$, $bla_{NDM}$ resistance genes in our samples. However, since these genes encompass diverse variants with variable resistance phenotypes, the presence of β-lactam ARGs does not necessarily indicate clinically significant resistance in all cases (50–52).

Tetracyclines are broad-spectrum antibiotics that are commonly used to treat and manage various infections. They act by inhibiting the 30S ribosomal subunit, thereby halting bacterial growth and replication (53). We identified several tetracycline resistance genes, including flavin-dependent monooxygenase [*tet(X)*], major facilitator superfamily efflux pump genes [*tet(G)*, *tet(C)*, *tet(39)*, *tet(A)*, *tet(L)*], and ribosomal protection protein genes [*tet(M)*, *tet(O)*, *tet(Q)*, *tet(W)*]. The WHO classifies tetracyclines as highly important antimicrobials, emphasizing the need for continuous surveillance to preserve their clinical efficacy (3).

Sulfonamides are used to treat a range of bacterial infections, including urinary tract infections, respiratory diseases, and opportunistic infections in immunocompromised patients (54). They act as competitive inhibitors of DHPS in the bacterial folate synthesis pathway, thereby inhibiting their growth and development (26). Specifically, the ARGs (*sul1*, *sul2*) were detected in our samples.

## Relative abundances of dominant ARGs are similar among regional neighbors and LMICs, with differences driven by low-abundance ARGs

Previous studies have demonstrated that ARG abundance profiles in wastewater are associated with geographic location and economic and development status (7, 55). To assess the similarities and differences between our samples and those from LMICs and regional neighbors, we compared our ARG abundance profiles with those derived from publicly available shotgun metagenomic data sets of untreated HWW. From the European Nucleotide Archive, we found data sets from previously published studies conducted in India, Burkina Faso, Benin, South Africa, China, and Singapore (6, 7, 12, 14, 56).

Figure 3 shows the drug-class-consolidated ARG relative abundances and the corresponding PCA plot. Our samples (PH) formed a relatively coherent cluster and were positioned closest to samples from India and China and were clearly separated from African samples (Burkina Faso [BFH], Benin [BH], and South Africa [SA]) along both PC1 and PC2. The drug classes with the highest contributions to PC1 were oxazolidinone, streptogramin, nitrofuran, and bicyclomycin-like, nucleoside, and phosphonic acid. Most of these are not among the drug classes with high relative abundance shown in Fig. 3A, suggesting that the differences are driven mostly by the low-abundance classes and that the dominant drug classes (macrolides, aminoglycosides, beta-lactam, tetracycline, sulfonamide) have similar profiles across the countries.

A global monitoring study of urban untreated sewage showed through a PCA of ARG concentration profiles that Asian and African samples clustered together and separated from European and North American samples (55). The separation was driven by mainly aminoglycosides, β-lactams, tetracyclines, and sulfonamides, while macrolide concentrations remained similar globally. It is interesting that our results from untreated HWW are consistent with their findings from urban sewage.

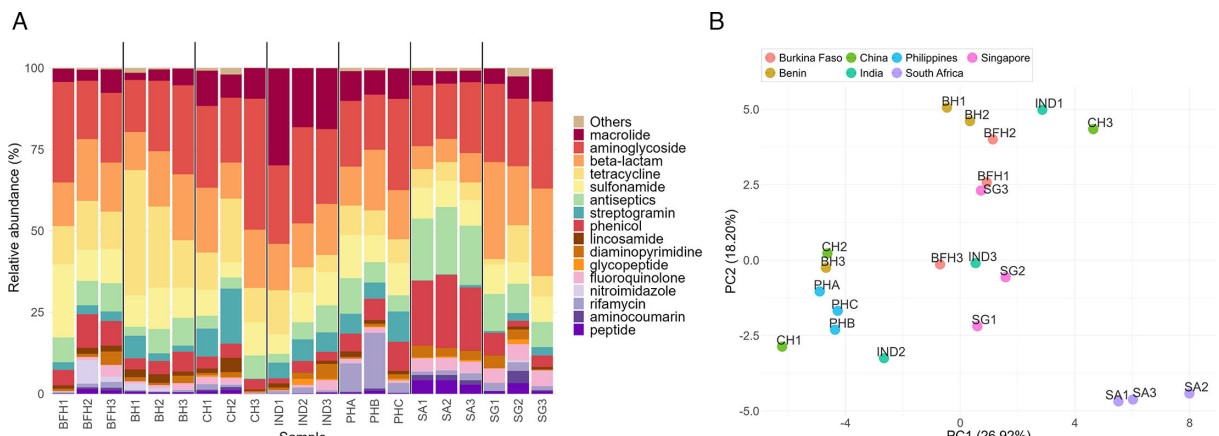

**FIG 3** Comparing Philippine samples to regional neighbors and LMICs (BFH: Burkina Faso, BH: Benin, CH: China, IND: India, SA: South Africa, SG: Singapore). (A) Relative abundances of ARGs consolidated by drug classes. The Philippine samples (PHA, PHB, and PHC) were aggregated at the hospital level by computing the compositional mean of each ARG across samples from the same hospital using the *mean.acomp* function in the R compositions package. Philippine hospitals are denoted as PHA (hospital A), PHB (Hospital B), and PHC (Hospital C). (B) Principal component analysis (PCA) based on the relative abundances shown in panel A.

The similarities between samples may reflect shared country contexts, including similarities in antimicrobial access, limited routine susceptibility testing, variability in regulatory enforcement, and antimicrobial stewardship practices (57–59).

## High-risk ARGs were observed in all samples

Previous studies have classified certain ARGs as being high risk based on their abundance, mobility, and host pathogenicity, which have been linked to antibiotic treatment failures in hospital settings (5, 60). They further classified the high-risk ARGs into four categories—highest priority, high priority, highly important, or important—based on their clinical relevance and alignment with the WHO list of medically important antimicrobials. Multiple high-risk ARGs were identified across the three hospitals as shown in Fig. 4.

Multiple β-lactam resistance genes classified as highest priority ARGs were detected across all hospitals. The presence of these ARGs may be attributed to the frequent clinical use of β-lactam antibiotics, which corresponds to the increasing resistance rates among Gram-negative bacilli isolates from Philippine hospitals (4). National surveillance studies have frequently reported variants of β-lactam resistance genes, such as $bla_{AIM}$, $bla_{OXA}$, and $bla_{TEM}$, whereas $bla_{NDM}$ and $bla_{KPC}$ have been detected in a few clinical isolates

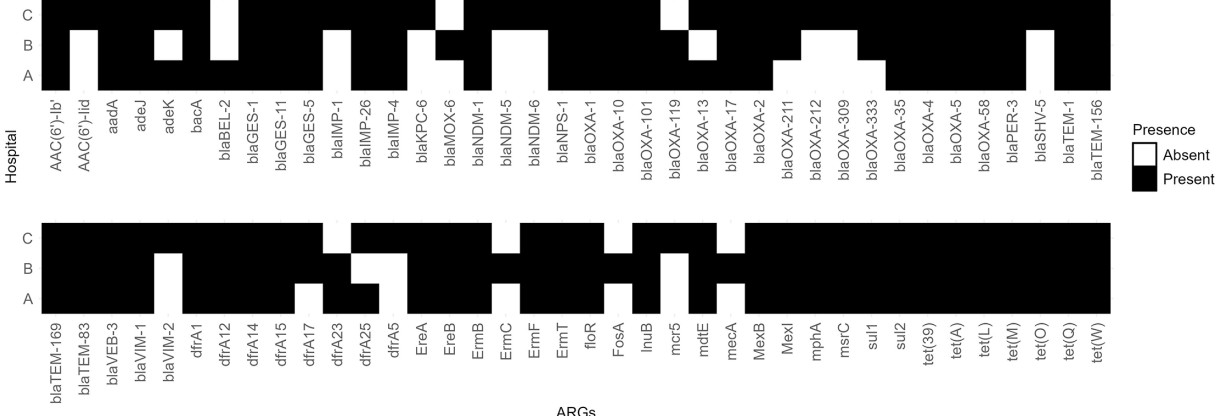

**FIG 4** High-risk ARGs. Black boxes indicate the presence of ARGs for each hospital A, B, and C.

(4). Of concern, $bla_{NDM}$, a carbapenemase that confers resistance to most β-lactam antibiotics, was first identified in India and has since spread globally (51). Similarly, $bla_{KPC}$, first identified from a *K. pneumoniae* isolate in the United States, has been documented in multiple countries including France and Israel (52, 61, 62). The detection of both $bla_{NDM}$ and $bla_{KPC}$ variants in our samples raises concern regarding the potential transmission of these high-risk ARGs and warrants further investigation at the national level.

The mobile colistin resistance gene (*mcr-5*) was also identified in our samples. Colistin serves as a last-resort treatment against multidrug-resistant Gram-negative infections and is classified as a highest priority critically important antimicrobial, highlighting the need to preserve its clinical effectivity (3, 63). Although previous clinical studies in the Philippines have primarily reported *mcr-1*, the detection of *mcr-5* in our study is noteworthy due to its higher resistance activity and greater potential for HGT, underscoring the need for continued surveillance of this resistance gene (4, 64).

Tetracycline ARGs, classified as highly important, were detected across all hospitals. We identified genes encoding major facilitator superfamily antibiotic efflux pumps [*tet(39)*, *tet(A)*, *tet(L)*] and tetracycline-resistant ribosomal protection proteins [*tet(M)*, *tet(O)*, *tet(Q)*, *tet(W)*]. Previous studies in the Philippines have reported the prevalence of ARGs *tet(A)*, *tet(C)*, *tet(M)*, and *tet(L)* in *E. coli* and *P. aeruginosa* isolates, highlighting the similarities of our results with national surveillance efforts (4, 65).

## Disinfectant resistance genes (*qacEΔ1* and *qacL*) were identified for the first time in Philippine HWW samples

To the best of our knowledge, this is the first report of *qacE*Δ1 and *qacL* genes in HWW from the Philippines. Both genes confer resistance to QACs, which are commonly used as disinfectants and cleaning agents (37). The increased use of QAC-containing products during the COVID-19 pandemic has resulted in greater release of these compounds into the environment (66, 67). The extensive use of QAC-based disinfectants in hospitals during the pandemic may have contributed to the high abundance of *qacEΔ1* and *qacL* resistance genes in our samples.

The presence of these genes in hospital wastewater has also been documented in China, Israel, Australia, and South Africa (6, 8, 40, 41, 68). The detection of disinfectant resistance genes in our samples underscores the need for further investigation in the Philippines to evaluate the potential risks of disinfectant resistance in both clinical and environmental contexts.

## Phyla Pseudomonadota, Actinomycetota, and Bacteroidota dominated the HWW microbial community

We assessed the bacterial taxonomic composition of the samples. Figures 5 and 6 show the relative abundances at the phylum and genus levels, respectively. Figure 7 shows a Sankey diagram illustrating the distribution of taxa in samples A1, B1, and C1.

At the phylum level, Pseudomonadota (synonym Proteobacteria), Actinomycetota (synonym Actinobacteria), and Bacteroidota (synonym Bacteroidetes) were the most abundant in all our samples as shown in Fig. 5 and 7. These were also reported to be most abundant in HWW samples from other LMICs, such as Burkina Faso, Benin, and India (7, 14), highlighting similarities in phylum-level bacterial composition to our HWW samples. Members of Actinomycetota and Bacteroidota are known to be the major constituents of the human gut microbiome and are present in wastewaters (12, 69). Members of Pseudomonadota are known human pathogens harboring various resistance genes against aminoglycosides, sulfonamides, β-lactams, and tetracyclines (70, 71). Under the phylum Pseudomonadota, some ESKAPEE pathogens were detected at relatively lower abundances in our samples. These pathogens are responsible for a wide range of human infections and include strains that exhibit resistance to most antibiotics, prompting their classification as "high-priority pathogens" by the WHO (72–76).

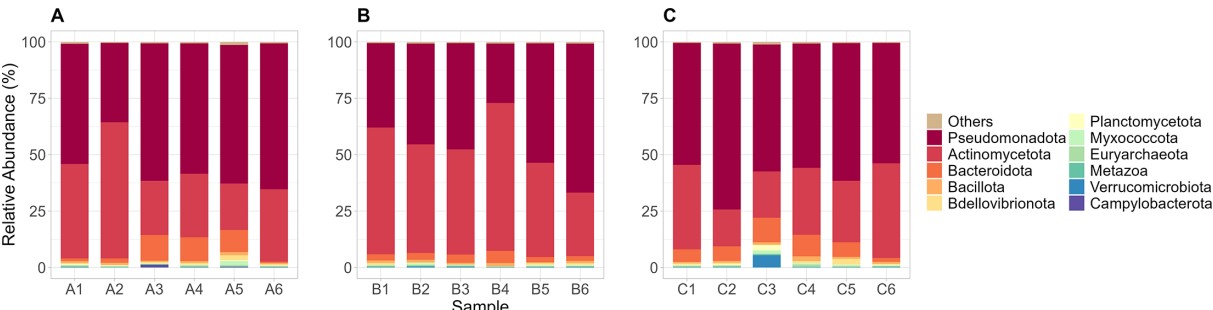

**FIG 5** Relative abundance of the top phyla from Hospital A (A), Hospital B (B), and Hospital C (C). Phyla not part of the top were grouped as "others."

At the genus level, *Rhodococcus*, *Stenotrophomonas*, and *Pseudomonas* were consistently among the most abundant across most samples (Fig. 6 and 7). *Rhodococcus* is typically found in soil, water, and animal hosts with a relatively low prevalence in hospital environments. Interestingly, in our study, this genus was detected at high abundance in HWW samples. Previous reports of human infections caused by several *Rhodococcus* species indicate their increasing clinical relevance and potential to harbor ARGs within hospital settings (77–79). Similarly, a study in China has detected the presence of *Rhodococcus* in HWWs (9), while other studies have identified them from patient samples (80, 81). *Stenotrophomonas* have been previously described as opportunistic pathogens commonly infecting humans, with known multidrug-resistant species, such as *Stenotrophomonas maltophilia* (82–85). This genus has been documented in various hospital plumbing systems, including sinks, showers, and toilets, which likely contributes to their presence in HWWs (86). Similarly, reports in India, Singapore, and the United Kingdom have also identified the genus in HWWs (12, 14, 87). *Pseudomonas* is a clinically significant opportunistic pathogen capable of causing a wide range of human infections in hospital settings. They are frequently reported to exhibit resistance to most antibiotics, especially against β-lactams (72, 73).

Interestingly, *Mycolicibacter* and *Pseudoxanthomonas* were also detected in our samples (Fig. 7). Although both genera are generally considered non-pathogenic to humans, previous studies have reported specific species with the potential to cause human infections and cause severe disease outcomes (88–90).

Other highly abundant genera included *Brevundimonas*, *Microbacterium*, *Mycolicibacterium*, *Nocardoides*, *Rhodanobacter*, and *Thermomonas* (Fig. 6 and 7). These genera have been previously described as components of HWW bacterial communities that shape the HWW environment and contribute to the removal of organic and inorganic compounds from wastewaters (7, 14, 87, 91, 92), which may partially explain their detection at high relative abundances in this study.

## Diverse MGEs were detected across all HWW samples

We identified signatures of a broad diversity of MGEs of IS and transposons, integrons, and plasmids. Among the most abundant IS and transposons (Fig. 8), we observed the presence of IS1071, ISPxasp1, IS1247, ISVsa3, ISAba1, and ISStma11. Several studies have reported these IS elements to carry ARGs conferring resistance to β-lactam, tetracyclines, aminoglycosides, sulfonamides, macrolides, and various disinfecting agents, and associated with genera, such as *Klebsiella*, *Pseudomonas*, *Stenotrophomonas*, *Rhodococcus*, and *Mycobacterium*, among others (93–96).

Class 1 integrons (intI1) were the most abundant type of integrons detected in our samples. Prior research has shown that these integrons were frequently associated with Gram-negative bacteria in clinical settings and commonly harbor ARGs conferring resistance to β-lactam, aminoglycosides, sulfonamides, and various disinfectants (97).

We have also identified a diversity of plasmid types that carry ARGs, including IncQ2 (NCBI accession no. FJ696404), IncP6 (NCBI accession no. JF785550), repUS43 (NCBI

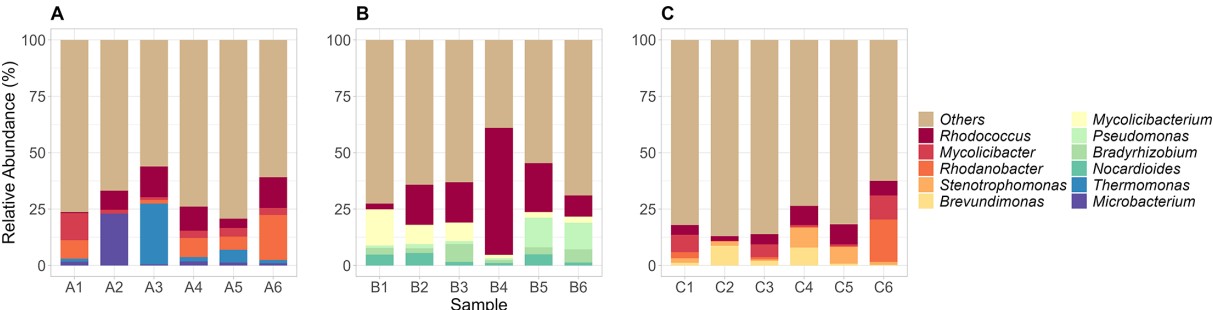

**FIG 6** Relative abundance of the top genera from Hospital A (A), Hospital B (B), and Hospital C (C). Genera not part of the top were grouped as "others."

accession no. CP003584), Col440II (NCBI accession no. CP023921), Col(pHAD28) (NCBI accession no. KU674895), and ColRNAI (NCBI accession no. DQ298019), among others (Fig. 9). Previous studies have reported that these plasmids can carry multiple ARGs conferring resistance to β-lactam, tetracyclines, carbapenems, macrolides, and various disinfectants, among others (97–99)

## Our data supplement the findings of the Philippine antibiotic resistance surveillance program

The Philippine Department of Health (DOH) established the Antimicrobial Resistance Surveillance Program (ARSP) in 1988 to monitor the prevalence and trends of antimicrobial resistance in selected bacterial pathogens. The ARSP reports phenotypic resistance rates of 11 species (*S. pneumoniae*, *H. influenzae*, *S. enterica*, *Shigella*, *V. cholerae*, *N. gonorrhoeae*, *S. aureus*, *Enterococcus*, *E. coli*, *K. pneumoniae*, *P. aeruginosa*, and *A. baumannii*) against commonly used antibiotics (4).

ARBs, such as *E. coli*, *K. pneumoniae*, *P. aeruginosa*, and *A. baumannii*, among others, showed resistance rates above 50% against drug classes of aminoglycosides, β-lactams, tetracyclines, and sulfonamides, among others (4). The detection of abundant ARGs in our samples conferring resistance to aminoglycosides, β-lactams, tetracyclines, and sulfonamides supplements ARSP reports by providing insights into genotypic resistance mechanisms.

The ARSP reports on clinically significant bacterial species that play a role in the spread of AMR in hospital settings. The presence of other genera such as *Rhodococcus*, *Mycolicibacter*, and *Pseudoxanthomonas* in our samples—isolates of many of which have been previously reported to harbor antimicrobial resistance (see, e.g., references 80,

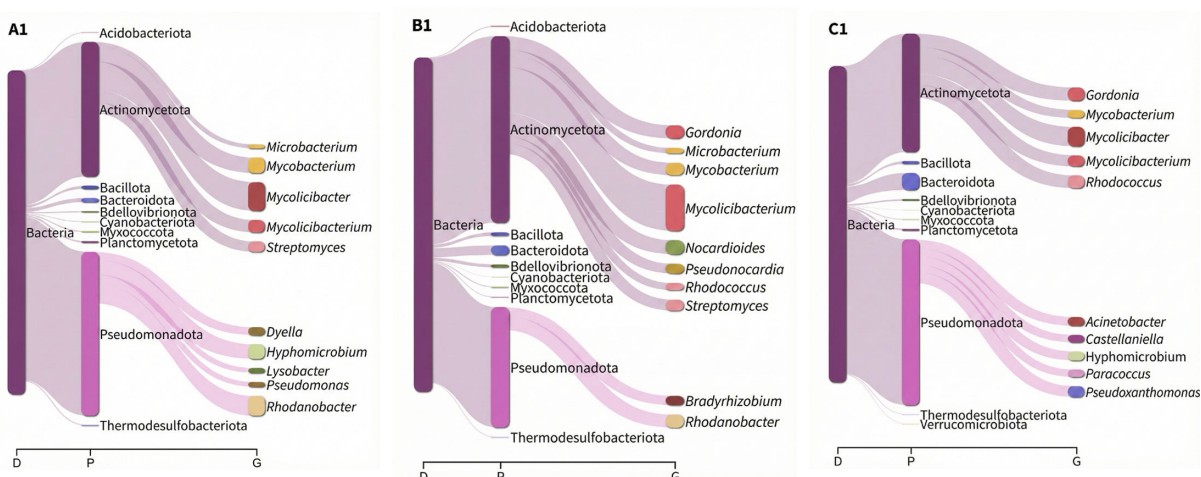

**FIG 7** Sankey diagram (generated using the R pavian package) illustrating the bacterial taxonomic distribution of samples A1, B1, and C1.

88, 90)—underscores the potential of shotgun metagenomics to identify underexplored bacterial taxa relevant for AMR surveillance.

## Our results suggest potential links between antimicrobial use and ARG profiles

Integrated surveillance studies from other countries have demonstrated associations between antimicrobial use (AMU) and resistome composition in hospital-associated environments using high-resolution metagenomic approaches (13, 100). In the Philippines, direct links between AMU and ARG profiles remain poorly defined, as existing studies have largely focused on prescription-based AMU assessments without complementary resistome analyses, limiting insight into how antibiotic consumption shapes environmental ARG patterns (101, 102). A multi-year assessment of a single private hospital reported a gradual increase in overall AMU, with β-lactams, macrolides, fluoroquinolones, and aminoglycosides accounting for the majority of prescribed antibiotics (102). Similarly, hospital-wide evaluation in another tertiary found the most frequently prescribed antibiotics for therapeutic use included penicillin in combination with a beta-lactamase inhibitor, piperacillin, cefuroxime, azithromycin, and ceftriaxone. Collectively, these studies indicate that β-lactams and macrolides were consistently among the most frequently prescribed antibiotic classes in Philippine hospitals. (101).

We observed a high abundance of ARGs associated with resistance to β-lactams and macrolides, which is consistent with the dominant antibiotic classes reported in limited Philippine AMU studies. While this suggests a potential relationship between antibiotic usage patterns and ARG profiles, the absence of an integrated AMU–ARG surveillance data in the country hinders robust correlation analyses.

## Conclusion

Our study investigated the diversity and abundance of ARGs, bacterial taxa, and MGEs in HWW in three hospitals in Metro Manila, Philippines. ARGs conferring resistance to critically important antibiotics including β-lactams, colistin, and tetracyclines were dominant in our samples. Our resistome profiles are broadly similar to those obtained from hospital wastewater samples of neighboring countries and other low-and-middle-income countries, suggesting there might be shared challenges in AMR management. Additionally, this is the first report of disinfectant resistance genes (*qacEΔ1*, *qacL*) in HWW in the Philippines. Genera, such as *Stenotrophomonas*, *Rhodococcus*, and *Pseudomonas*, were detected at high relative abundances across all sampling sites. MGEs that were previously reported to harbor clinically important drug classes were also detected.

This exploratory study has several limitations. The sample size is insufficient for comprehensive surveillance, and the limited temporal coverage does not allow for the assessment of seasonal variability. In addition, the average sequencing depth of approximately 8.8 Gbp per sample constrains the detection of low-abundance taxa and rare antimicrobial resistance genes within the highly complex hospital wastewater metagenomes. The short read length (150 bp) further restricts more detailed analyses,

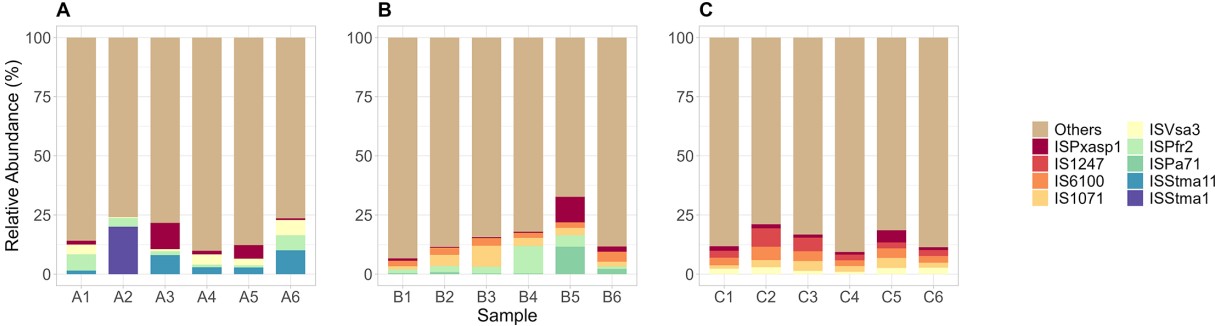

**FIG 8** The relative abundance of the top IS from Hospital A (A), Hospital B (B), and Hospital C (C). IS not part of the top and were grouped as "others."

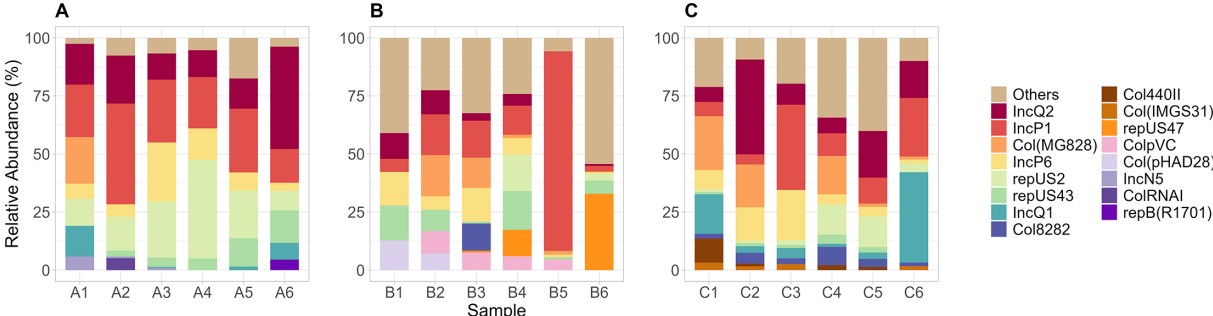

**FIG 9** The relative abundance of the top plasmid replicons from Hospital A (A), Hospital B (B), and Hospital C (C). Plasmid replicons not part of the top were grouped as "others."

including the identification of variants of resistance genes—some of which may not confer resistance—as well as the establishment of associations between ARGs and mobile genetic elements. It should also be noted that this study collects only DNA, which does not necessarily indicate gene expression or phenotypic resistance, and therefore does not directly reflect the associated clinical risks. Finally, the absence of antimicrobial usage data precluded analyses linking antimicrobial consumption patterns to observed ARG abundances, which would have provided additional insights to inform antimicrobial resistance surveillance and management strategies in the Philippines.

Despite these limitations, this first metagenomic profiling of hospital wastewater in the Philippines provides a valuable preliminary evidence base for future antimicrobial resistance and epidemiological surveillance efforts, particularly those leveraging wastewater-based approaches.

### ACKNOWLEDGMENTS

The authors express their gratitude to Dr. Evelina Lagamayo, Ms. Sherill Tesalona, and Mr. Joemarie Malana for their valuable assistance in obtaining ethics clearances for the three hospitals. We also extend our thanks to Mr. Elmer Abanzado, Engr. Sherwin Mina, and Mr. Emerson Vergara for their support in the collection of samples. We also acknowledge the members of the e-ASIA JRP group for their insights and constructive discussions that greatly contributed to this study.

Daphne Go is thankful to the Department of Science and Technology and the Engineering Research and Development for Technology (ERDT) scholarship program for funding her bachelor of science and master of science in computer science. This research was funded by the Department of Science and Technology Philippine Council for Health Research and Development (DOST-PCHRD) through East Asia Science and Innovation Area Joint Research Program (e-Asia JRP). The sponsors had no role in study design, data collection and analysis, decision to publish, or manuscript preparation.

### AUTHOR AFFILIATIONS

[1]Department of Biology, College of Science, De La Salle University, Manila, Philippines
[2]Systems & Computational Biology Unit, Center for Natural Sciences and Environmental Research, De La Salle University, Manila, Philippines
[3]Department of Software Technology, College of Computer Studies, De La Salle University, Manila, Philippines
[4]Bioinformatics Laboratory, Advanced Research Institute for Informatics, Computing, and Networking, De La Salle University, Manila, Philippines
[5]Institute of Arts and Sciences, Far Eastern University, Manila, Philippines
[6]Bioinformatics and Genomics Program, Pennsylvania State University College of Medicine, Hershey, Pennsylvania, USA

[7]Department of Mathematics and Statistics, College of Science, De La Salle University, Manila, Philippines

[8]Research & Biotechnology Division, St Luke's Medical Center, Quezon City, Philippines

## AUTHOR ORCIDs

Paul K. Yu ⓘ http://orcid.org/0000-0001-7848-5050
Angelyn R. Lao ⓘ http://orcid.org/0000-0003-3028-9610
Ma. Luisa D. Enriquez ⓘ http://orcid.org/0000-0002-4542-3627
Llewelyn M. Espiritu ⓘ http://orcid.org/0000-0002-5129-2635
Anish M. S. Shrestha ⓘ http://orcid.org/0000-0002-9192-9709

## FUNDING

| Funder | Grant(s) | Author(s) |
| --- | --- | --- |
| Philippine Council for Health Research and Development | e-Asia JRP | Angelyn R. Lao |

## AUTHOR CONTRIBUTIONS

Jiaan Carlo E. Santos, Investigation, Methodology, Writing – original draft, Writing – review and editing | Daphne Janelyn L. Go, Investigation, Methodology, Software, Visualization | Robert D. Unciano, Investigation, Methodology, Project administration, Resources | Paul K. Yu, Formal analysis, Investigation, Methodology, Software | Angelyn R. Lao, Conceptualization, Funding acquisition, Project administration, Supervision | Ma. Luisa D. Enriquez, Conceptualization, Funding acquisition, Resources, Supervision | Llewelyn M. Espiritu, Conceptualization, Funding acquisition, Methodology, Supervision, Writing – review and editing | Anish M. S. Shrestha, Conceptualization, Formal analysis, Methodology, Supervision, Writing – original draft, Writing – review and editing

## DATA AVAILABILITY

The data sets generated during the current study are available in the DDBJ Sequence Read Archive under the accession number PRJDB37674. The computational pipeline used for data analysis is available at the following URL: https://github.com/bioin-fodlsu/attack_amr_pipeline.

## ADDITIONAL FILES

The following material is available online.

Open Peer Review

**PEER REVIEW HISTORY (review-history.pdf).** An accounting of the reviewer comments and feedback.

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
