## [Reviewer comments · Microbiology Spectrum]

Microbiology Spectrum

Investigating the resistome, taxonomic composition, and mobilome of bacterial communities in hospital wastewaters of Metro Manila using a shotgun metagenomics approach

Jiaan Santos, Daphne Go, Robert Unciano, Paul Yu, Angelyn Lao, Ma. Enriquez, Llewelyn Espiritu, and Anish Shrestha

Corresponding Author(s): Anish Shrestha, De La Salle University

Review Timeline:

Submission Date:	February 10, 2026
Editorial Decision:	February 11, 2026
Revision Received:	February 12, 2026
Accepted:	February 19, 2026

Editor: Blaire Steven

Reviewer(s): The reviewers have opted to remain anonymous.

Transaction Report:

DOI: <https://doi.org/10.1128/spectrum.03963-25>

Re: Spectrum03963-25 (Investigating the resistome, taxonomic composition, and mobilome of bacterial communities in hospital wastewaters of Metro Manila using a shotgun metagenomics approach)

Dear Dr. Anish Shrestha:

Thank you for the privilege of reviewing your work. Below you will find my comments, instructions from the Spectrum editorial office, and the reviewer comments.

After having read the manuscript and the authors response to reviewers comments I am happy to recommend editorial acceptance of the study.

I am pleased to inform you that your manuscript has been editorially accepted for publication. However, there are a few additional questions in the submission form that need to be answered before the final decision. Once these are completed, please return your submission so that I can move your paper forward to acceptance.

Revision Guidelines

Sincerely,
Blair Steven
Editor
Microbiology Spectrum

Response to reviewers' comments

We thank the editors for allowing us to transfer our manuscript to the more suitable *Microbiology Spectrum*. We are grateful to the reviewers of our submission to *Applied and Environmental Microbiology* for their careful evaluation of the manuscript and for the numerous constructive and insightful comments. Based on their comments, we have substantially revised the manuscript to incorporate additional experiments and evaluations recommended by the reviewers. Changes in the manuscript are marked in red. We provide a point-by-point response to the reviewers' comments below.

Reviewer comment	Our remarks
Reviewer 1	
There is no clear objective statements. It is unclear what hypotheses or knowledge gaps were addressed.	This is an exploratory study, and not a hypothesis-driven one. Our objective is to profile the antibiotic resistance gene abundance profile, bacterial community composition, and the occurrence of MGEs in untreated hospital wastewater (HWW) from tertiary hospitals in Metro Manila, Philippines. To the best of our knowledge, this work represents the first metagenomics-based profiling of HWW in the Philippines and establishes an initial evidence base on the diversity and prevalence of ARGs and clinically relevant pathogens in healthcare-associated wastewater. We have rephrased the last paragraph of the Introduction section to make our objectives clearer.
In order to check validity and utility of the results, it is expected to compare with in-hospital surveillance. Even if the in-hospital surveillance data in the target hospitals are not available, it should be compared with national clinical surveillance.	We agree that comparing our findings with in-hospital surveillance data would strengthen the validity and utility of the results. However, in-hospital surveillance data for the target hospitals were not available to the authors at the time of this study. Additionally, national clinical surveillance data in the Philippines are limited and largely accessible through the Antimicrobial Resistance Surveillance Program

	(ARSP) reports. In Section 3.8, we have incorporated relevant ARSP findings into the discussion and highlighted similarities between our resistome profiles and the increasing AMR trends reported by the national surveillance system.
Abundant ARGs in clinical ARB are known to vary by country. It is expected to deepen the discussion based on the clinical surveillance and/or past studies on abundant ARGs in clinical ARB.	We have expanded the Results and Discussion section to include a comparative analysis of publicly available hospital wastewater resistome data from regional neighbors and low- and middle-income (LMIC) countries. We observed that the relative abundances of dominant ARGs are similar among regional neighbors and LMICs, with differences driven by low-abundance ARGs. Please see the new Section 3.3 for details.
Impacts of antimicrobial use (AMU) in hospitals on AMR are discussed. However, AMU data in target hospitals or statistical AMU data in Philippines are not presented. Presentation of such data and quantitative comparison, i.e. statistical test, multivariate analyses, with them are necessary to support their discussion.	We agree that quantitative AMU data and integrated AMU–ARG analyses are necessary to robustly support discussions on antimicrobial resistance. However, AMU data from the study sites as well as nationally integrated AMU–ARG public datasets are currently limited in the Philippine context. We were able to obtain just 2 studies that have identified AMU based on prescribed antibiotics to the patients. We therefore framed our discussion qualitatively – rather than quantitatively – referencing the two Philippine-based AMU studies. We have added Section 3.9 to include this discussion and to acknowledge our limitations.
Reviewer 2	
The authors should provide more detailed description of the 3 hospitals, for example what type of hospitals, what is the daily number of patients/bed capacities per hospital, antibiotic usage, cleaning protocols inside the hospitals etc. This would provide	We have added to the description of the hospital sites. Please see Section 2.1 Table 1. Unfortunately, we were not able to obtain precise information about the treatment protocol at each hospital except for the disclosure that each

some context to the type of ARGs and MGEs that are detected in wastewater.	hospital employs its own treatment process, which includes physicochemical and biological methods, before discharging the wastewater into the city sewage system.
The samples were collected biweekly over a 3-month period. This means that about 6 samples were collected per hospital which is a rather small sample size and short duration for a surveillance study	We agree that six samples per hospital over a three-month period is not sufficient for an extensive surveillance study. However, the primary objective of this work was descriptive profiling of hospital wastewater (HWW) resistomes, bacterial community composition, and mobilomes in the Philippine setting rather than conducting a surveillance program. We have revised the last paragraph in the Introduction section to clarify our objective. Additionally, a new second paragraph in the Conclusion section touches on the limitations of the study, including the one about sample size raised here.
Sequencing reads should be deposited in public repositories.	The raw sequencing reads have already been deposited in the DNA Data Bank of Japan (DDBJ) Sequence Read Archive under accession number PRJDB37674 and are currently under a Hold-Until-Published arrangement. This was stated in Section 7 Availability of data and materials of the original draft.
The authors may be overinterpreting the detection of those ARGs and overemphasizing the potential detrimental impact on public health without an actual context on whether these genes are functionally expressible. Detection by means of DNA-based does not necessarily equate to risks. In addition, without any background context related to the hospitals (point 1 raised above), it is difficult to utilize the findings for	We agree that we collect only DNA data, which does not necessarily indicate gene expression or phenotypic resistance, and therefore does not directly reflect the associated clinical risks. This is a general limitation of metagenomics-based studies. We make note of this limitation in the last paragraph of Section 3.1 and also

intervention measures (thus diminishing the impact/contribution of this study)	in the second paragraph of the Conclusion section.
The authors can perhaps strengthen the study by doing a meta-data analysis (downloading global datasets of ARGs from hospital wastewaters) and compare them to determine the differences and similarities.	We expanded the Results and Discussion section to include a comparative analysis of publicly available hospital wastewater resistome data from regional neighbors and low- and middle-income (LMIC) countries. We observed that the relative abundances of dominant ARGs are similar among regional neighbors and LMICs, with differences driven by low-abundance ARGs. Please see the new Section 3.3 for details.
Reviewer 3	
The aims of the work are not clearly articulated, and the research conducted is not hypothesis-driven: The manuscript needs a more precise statement of its aims and contributions of the findings.	This is an exploratory study, and not hypothesis-driven. Our objective is to characterize the antibiotic resistance gene abundance profile, bacterial community composition, and the occurrence of MGEs in untreated hospital wastewater (HWW) from tertiary hospitals in Metro Manila, Philippines. To the best of our knowledge, this work represents the first metagenomics-based profiling of HWW in the Philippines and establishes an initial evidence base on the diversity and prevalence of ARGs and clinically relevant pathogens in healthcare-associated wastewater. We have rephrased the last paragraph of the Introduction section to make our objectives clearer.
a. Although the sample collection strategy consisting of spatial as well as temporal replicates stands as a great merit of the study, it is not clearly stated which research question the choice of sampling strategy aims to match. Perhaps the idea was to examine	Sampling was affected by budgetary issues, and although we did sample at various time points, they do not cover a large enough time interval to see any seasonality effects. In this paper, we have ignored the temporal aspect.

those ARG-carrying bacteria that are not screened by cultivation by the national Antimicrobial Resistance Surveillance Program (ARSP) for the clinical specimens mentioned, to gain a better understanding of the resistance level carried by the human population? But if so, it is unclear why the authors seek the cultivable ESKAPEE taxa from the metagenomic data? Cultivation-based methods - performed by the local surveillance program - are more suitable for that.	We agree that cultivation-based approaches, such as those employed by the national ARSP, are more suitable for detailed analyses of cultivable clinical pathogens and their resistance phenotypes. We did not actively target or search for ESKAPEE taxa. Their detection was reported to highlight their presence within the HWW resistome context rather than to infer clinical relevance or transmission dynamics.
b. It could be highlighted within the aims that the hospital wastewater, as the chosen study material, will provide insights into the resistance burden of hospital patients (somehow ill people) rather than the whole community level (asymptomatic carriage of ARGs), which would require communal sewage sampling.	We agree that the scope of inference associated with HWW should be clearly articulated. In response, we have revised the introduction(second-to-last paragraph) to provide clearer context that HWW reflects hospital-related activities, including hospital-associated resistance pressures that may provide insights of the AMR burden within the hospital environment.
c. Description of the hospitals sampled should be added: E.g., What is the number of patients in the hospitals? What kinds of medical departments do they have (e.g., do they have infectious diseases wards, etc.)? Are they day or night hospitals? Private or public hospitals? Why were these hospitals chosen? To be able to evaluate the risk of further spread of these ARGs, the following question could be added to the manuscript if the authors think that they are relevant regarding the main scope of the research: Will the HWW from these hospitals be treated? How? Will the treatment reduce the risk of further spread of the antimicrobial resistance genes (ARGs)?	We have added to the description of the hospital sites. Please see Section 2.1 Table 1. Unfortunately, we were not able to obtain precise information about the treatment protocol at each hospital except for the disclosure that each hospital employs its own treatment process, which includes physicochemical and biological methods, before discharging the wastewater into the city sewage system.
d. Based on how the results are presented in the manuscript, the focus of the study seems to be the ARGs and MGEs of 'clinical concern'. However, the authors need to explicitly define the schemes based on what they categorize those already in the	We agree that the term “ARGs of clinical concern” were not sufficiently defined in the original manuscript. We now refer to them as “high-risk ARGs” as defined and classified by previous papers

Introduction section. For instance, the selection criteria for the genes in Figure 3 are not explained.	(doi.org/10.1038/s41467-021-25096-3, doi.org/10.1016/j.jhazmat.2024.133790) We have updated Section 3.4 to reflect these changes.
e. In general, the methodology used in the study should be justified in the manuscript based on the study focus. For instance, why was the CARD database among many other options selected to describe the resistomes in these samples and answer the research questions of the study?	There are indeed many ARG databases available. We selected the CARD as the reference ARG database due to its extensive coverage of resistance determinants and associated antibiotics, and its structured Antibiotic Resistance Ontology (ARO) for drug-class annotation. While it would not be practical to evaluate all the available databases, we had assessed ResFinder as an alternative option, but observed that it yielded fewer mappings/alignments compared to CARD for our samples.
2. There seems to be incorrectly executed data analysis for ARG detection and reporting using CARD database. a. Based on the reported scripts in (https://github.com/bioinfodlsu/attack_amr_pipeline/tree/main) and the results the nucleotide metagenomic reads have been used as such for mapping them against the entire CARD database, including housekeeping genes that require individual investigation for mutation-induced resistance, as well as genes from the CARD protein homolog model. The latter model is designed for sequences translated to amino acids. For these reasons, the results derived from this analysis cannot be considered reliable. Instead, the suggested workflow with CARD and metagenomic nucleotide reads as input is done by Resistance Gene Identifier (RGI). Moreover, the interpretation of the resulting hits must be carried out carefully: RGI will output hits categorized into Perfect, Strict, and Loose based on how well the sequences	We used Bowtie2 to map the metagenomic reads against the nucleotide version of CARD's Protein Homolog Model, specifically the CARD nucleotide reference sequences provided in the nucleotide_fasta_protein_homolog_model.fasta file of the downloaded CARD database. This strategy is consistent with the workflow provided by RGI through its rgi bwt mode, which allows short DNA sequences in FASTQ format to be aligned using Bowtie2, BWA, or KMA against nucleotide versions of CARD's protein homolog models. In accordance with the default RGI-bwt configuration, we restricted our analysis to the canonical curated CARD reference sequences by using only the nucleotide_fasta_protein_homolog_model.fasta file, and did not include

match, based on which the hits can be further filtered. I suggest that the authors examine the instructions related to CARD usage and result interpretation and rerun the ARG detection analysis according to the manual: https://github.com/arpcard/rgi/blob/master/docs/rgi_bwt.rst.	resistome or variant-based databases that are intended for mutation-induced resistance detection. As such, our analysis follows the same read-mapping strategy implemented by RGI-bwt and is intended for the detection and quantification of known ARGs based on nucleotide similarity. We have updated Section 2.5 to include these details. As a side note, existing studies have used a similar methodology (https://doi.org/10.1016/j.envpol.2023.121539, https://doi.org/10.1016/j.jece.2022.109216, https://doi.org/10.1111/lam.12842).
b. The authors have normalized the mapped ARGs to the average genome size. Given the huge variation of bacterial genomes, especially in highly diverse wastewater samples, this approach would result in a very broad estimate of the actual ARG/bacterial cell ratio. Instead, I would suggest more precise and commonly used approaches for ARG normalization, such as those relying on the detection of the 16S rRNA gene. A similar approach could be applied with the MGEs.	There are indeed several units for quantifying ARGs, each with its merits and demerits. Here we used ARG copy per genome (which is equivalent to ARG copy per cell, assuming one genome per cell, which is the unit recommended for metagenomics by this benchmarking study https://doi.org/10.1021/acs.est.3c00159). It is easier to interpret than ARG per 16S rRNA, since there could be multiple copies of rRNA genes per genome. It is also easier to compute than ARG per 16S rRNA as the latter requires additional mapping to 16s rRNA databases. More importantly though, since our data is compositional, we only report relative abundances and not absolute abundances. As far as computing relative abundances is concerned, both ARG copy per genome and ARG copy per rRNA gene yield exactly the same result.
c. With the reported sequencing depth (8.8 Gb of data/sample), the ability to detect rare	We agree that the reported sequencing depth may limit the detection of

species and genes - that is, the ARGs and ARG-carrying bacteria here - particularly in complex wastewater metagenomes, is likely limited. This could be acknowledged as a limitation of the study in the manuscript.	low-abundance species and rare ARGs in complex hospital wastewater metagenomes. We mention this along with other limitations in the second paragraph of the Conclusions section.
3. The interpretation of results draws overly strong conclusions from limited evidence. a. In the abstract and later in the Results and Discussion section, the authors state that the study investigates antibiotic-resistant bacteria (ARBs). However, the shotgun metagenomic data and analysis presented in the manuscript do not allow drawing such conclusions (identification of the ARG-carrying bacteria). So, in fact, within the limitations of this study, nothing can be said about the bacteria carrying ARGs here. This should be highlighted in the text. However, the authors seem to assume that the high abundance of certain taxa in HWW means that they are the likely hosts for the ARGs: "The presence of other ARBs such as Rhodococcus, Mycolicibacter, and Pseudoxanthomonas in our samples underscores the potential of shotgun metagenomics to identify underexplored ARBs for AMR surveillance. These organisms may also contribute to AMR dissemination and represent possible emerging pathogens warranting further investigation." However, quite the opposite is more likely, as the ARG-carrying pathogens are typically among the rare taxa in the highly complex wastewater communities, and more importantly, the linkage between ARG and its carrier cannot be established based on this data. Therefore, the interpretation is highly speculative, not based on any evidence, and should be revised.	Indeed, that certain taxa were found in high abundance does not mean they are ARBs. We have modified the problematic sentence in Section 3.8.
b. The general interpretation of the results lacks precision and is rather speculative without evidence base from the research presented in the manuscript. For instance: "Both ARGs are typically plasmid-encoded, enhancing their potential for HGT and environmental dissemination". Similarly, the	Indeed the association of MGEs and ARGs was not established by this study. We have revised the previously ambiguous sentences to explicitly clarify when interpretations are based on prior studies rather than on our own findings. The updated text appears in

authors conclude that: "MGEs associated with clinically important drug classes were also detected." but the research conducted in the manuscript does not provide any evidence on the association of MGEs to any ARGs, if that is what they meant by the sentence. The authors should take care to clearly distinguish the findings of this study from those of previous publications.	Section 3.1 and Section 4 ("MGEs associated...".)
Minor - The authors could consider revising the manuscript title to better convey the main focus or highlight the research outcomes.	We have purposely chosen a broad title since this is an exploratory study with no specific hypothesis being tested, and we report on a wide range of results regarding the ARG, taxonomic, and MGE profile of the hospital waste waters sampled.
-The sample details in Table 1 should be organized so that there is one sample replicate per row. Also, additional sample metadata could be included.	We have reorganized Table 1 so that each sample replicate is presented in a separate row, improving clarity and traceability of the sampling design.
-In the abstract, the mixed use of the words "disinfectant" and "antiseptic" resistance genes for the qac genes is confusing. The authors should use a single term consistently throughout the manuscript.	We have edited the manuscript to use the term "disinfectant" to minimize confusion.
- The authors have included parts of their results in the Introduction section, while this section should be free from results. Moreover, the statements there, like "Our findings demonstrate that bacterial composition and ARG profiles in Philippine tertiary hospitals are similar to those observed globally." are too vague and strong, especially without any references to previous studies, and should be excluded or rephrased.	We have removed the result highlights from the Introduction section.
- The Introduction contains several statements that lack references.	We have added citations to several statements that may have lacked them previously. Please see the sentences marked in red in the first three paragraphs in the Introduction section.

- The authors could consider adding a description or discussion on how the results in the bar plot figures compare between the different hospitals and sample points.	We did not attempt to make statistical comparisons among the different hospitals, mainly because all the hospitals serve the same metropolis and also serve as national referral centers. Even if there were statistical differences, it would not be possible to interpret those differences, since we were not able to collect information on various influencing factors like AMU usage, cleaning protocols, etc.
- Part of the results showing less impactful results (e.g., the antibiotic classes (ARO)) could be excluded or transferred to supplementary material. Moreover, the authors could explore other visualization strategies instead of focusing solely on the most abundant ARGs or taxa, which might overlook more insightful trends.	We have decided to retain the results on antibiotic classes in the main manuscript since it connects to new Section 3.3 where we compare our ARG profile against other countries, and refer to articles that also use the drug class abundance to compare global samples. As an alternative visualization technique, we have added a new Fig 7 that displays the taxonomic distribution using a Sankey diagram.
- It is unclear whether the different MGE databases are dereplicated for redundant sequences or if the mappings are run separately for each database.	The mappings are run separately for each database.
- The colors in Figure 5 are not well distinguishable.	We have reduced the number of genera to make the colors more distinguishable. Please see new Figure 6.
- The description of the detected ARG variants lacks precision: For example, detection of blaOXA genes is mentioned, but as there is a great variety of resistance and non-resistance phenotypes associated with different variants of blaOXA genes, this should be articulated in the text. Also, for instance sul4 is mentioned in the text as one of the abundant ARGs, but it is not shown in any of the figures. The authors should clearly indicate where this result can be found.	We have addressed these concerns by improving precision in the description of detected ARG variants. Specifically, we clarified the discussion of blaOXA genes by noting the diversity of blaOXA variants and their variable resistance phenotypes, and we revised the text to avoid overgeneralization. See section 3.2. Since this is a general limitation of this study, we mention this also at the end of Section 3.1 and in the

- Mycolicibacter and Pseudoxanthomonas are referred to as 'environmental contaminants', but it is unclear what this means. The authors should rephrase or exclude this definition.

Conclusion section where we discuss the limitations.

Additionally, we removed the description of *Mycolicibacter* and *Pseudoxanthomonas* as "environmental contaminants" to avoid potential confusion. We revised the text to clarify that these genera are generally considered non-pathogenic to humans, while acknowledging that recent studies have reported certain species within these genera with pathogenic potential. see section 3.6.

Re: Spectrum03963-25R1 (Investigating the resistome, taxonomic composition, and mobilome of bacterial communities in hospital wastewaters of Metro Manila using a shotgun metagenomics approach)

Dear Dr. Anish Shrestha:

I am happy to move this manuscript forward to publication.

Your manuscript has been accepted, and I am forwarding it to the ASM production staff for publication. Your paper will first be checked to make sure all elements meet the technical requirements. ASM staff will contact you if anything needs to be revised before copyediting and production can begin. Otherwise, you will be notified when your proofs are ready to be viewed.

Sincerely,
Blair Steven
Editor
Microbiology Spectrum